# Pathomechanism Characterization and Potential Therapeutics Identification for Parkinson’s Disease Targeting Neuroinflammation

**DOI:** 10.3390/ijms22031062

**Published:** 2021-01-21

**Authors:** Chiung-Mei Chen, Chien-Yu Yen, Wan-Ling Chen, Chih-Hsin Lin, Yih-Ru Wu, Kuo-Hsuan Chang, Guey-Jen Lee-Chen

**Affiliations:** 1Department of Neurology, Chang-Gung Memorial Hospital, Chang-Gung University College of Medicine, Taoyuan 33302, Taiwan; cmchen@adm.cgmh.org.tw (C.-M.C.); chli416@hotmail.com (W.-L.C.); yihruwu@adm.cgmh.org.tw (C.-H.L.); gophy5128@adm.cgmh.org.tw (Y.-R.W.); michelle8237@hotmail.com (K.-H.C.); 2Department of Life Science, National Taiwan Normal University, Taipei 11677, Taiwan; s0951352@alumni.ncyu.edu.tw

**Keywords:** Parkinson’s disease/α-synuclein, NLRP1/3, IL-1β/IL-6, IkBα/P65, JNK/JUN, P38/STAT1, JAK2/STAT3, therapeutics

## Abstract

Parkinson’s disease (PD) is a common neurodegenerative disorder characterized by the loss of dopaminergic (DAergic) neurons and the presence of α-synuclein-containing Lewy bodies. The unstructured α-synuclein forms insoluble fibrils and aggregates that result in increased reactive oxygen species (ROS) and cellular toxicity in PD. Neuroinflammation engaged by microglia actively contributes to the pathogenesis of PD. In this study, we showed that VB-037 (a quinoline compound), glycyrrhetic acid (a pentacyclic triterpenoid), *Glycyrrhiza inflata* (*G. inflata*, a Chinese herbal medicine), and Shaoyao Gancao Tang (SG-Tang, a formulated Chinese medicine) suppressed the nitric oxide (NO) production and interleukin (IL)-1β maturation in α-synuclein-stimulated BV-2 cells. Mouse inflammation antibody array further revealed increased IL-1α, IL-1β, tumor necrosis factor (TNF)-α, interferon (IFN)-γ, IL-6, granulocyte-macrophage colony-stimulating factor (GM-CSF) and granulocyte colony-stimulating factor (G-CSF) expression in α-synuclein-inflamed BV-2 cells and compound pretreatment effectively reduced the expression and release of these pro-inflammatory mediators. The test compounds and herbal medicines further reduced α-synuclein aggregation and associated oxidative stress, and protected cells against α-synuclein-induced neurotoxicity by downregulating NLR family pyrin domain containing 1 (NLRP1) and 3 (NLRP3), caspase 1, IL-1β, IL-6, and associated nuclear factor (NF)-κB inhibitor alpha (IκBα)/NF-κB P65 subunit (P65), c-Jun N-terminal kinase (JNK)/proto-oncogene c-Jun (JUN), mitogen-activated protein kinase 14 (P38)/signal transducer and activator of transcription 1 (STAT1) and Janus kinase 2 (JAK2)/signal transducer and activator of transcription 3 (STAT3) pathways in dopaminergic neurons derived from α-synuclein-expressing SH-SY5Y cells. Our findings indicate the potential of VB-037, glycyrrhetic acid, *G. inflata,* and SG-Tang through mitigating α-synuclein-stimulated neuroinflammation in PD, as new drug candidates for PD treatment.

## 1. Introduction

Parkinson’s disease (PD), one of the most common neurodegenerative disorders, has the clinical features of resting tremor, rigidity, bradykinesia, and postural instability. The pathology is characterized mainly by progressive loss of the nigro-striatal dopaminergic neurons and the presence of cytoplasmic inclusion bodies (Lewy bodies) containing α-synuclein in the ventral midbrain [1]. Alpha-synuclein (*SNCA*), parkin RBR E3 ubiquitin protein ligase (*PRKN*), parkinsonism associated deglycase (*DJ1*), PTEN induced kinase 1 (*PINK1*), leucine rich repeat kinase 2 (*LRRK2*), ATPase cation transporting 13A2 (*ATP13A2*), VPS35 retromer complex component (*VPS35*), eukaryotic translation initiation factor 4 gamma 1 (*EIF4G1*), phospholipase A2 group VI (*PLA2G6*), F-box protein 7 (*FBXO7*), synaptojanin 1 (*SYNJ1*) and DnaJ heat shock protein family (Hsp40) member C6 (*DNAJC6*) have been identified to be the causative genes for familiar and early-onset PD (EOPD) [2]. Owing to the discovery of the causative genes, several pathogenic pathways were identified, which includes accumulation of aberrant or misfolded proteins, mitochondrial dysfunction, increased oxidative stress, impaired ubiquitin-proteasome function, failure of autophagy-lysosome and mitophagy, deficits in endosomal trafficking and inflammation (see review in [3]). Several genome wide-association studies (GWAS) also have identified novel genetic associations with PD and these genes are linked to the previously known pathogenic pathways as well as other less recognized pathways, such as endocytosis, transcriptional dysregulation, inflammation, and cytokine-mediated signaling [4,5].

PD brain at post-mortem has shown CD8^+^ and CD4^+^ T-cell infiltration and accumulations of microglia cells and astrocytes in substantia nigra [6]. Substantial evidence has also shown that microglial activation, nuclear factor kappa B (NF-κB) induced neuroinflammation and release of inflammatory factors may play an important role in the neurodegeneration of PD [7,8,9]. Aggregated α-synuclein could activate microglia, which leads to disease progression of PD [10]. Direct injection of α-synuclein into the substantia nigra resulted in the upregulation of mRNA expression of proinflammatory cytokines, the expression of endothelial markers of inflammation and microglial activation [11]. Neuroinflammation is also likely to play a key role in propagation of misfolded α-synuclein in a “prion-like” fashion in PD [12]. Gut microbiota promotes α-synuclein-dependent microglia activation, leading to neuroinflammatory damage in brains of mice [13]. Furthermore, several studies have suggested protective effects of anti-inflammatory drugs for PD animal models and epidemiological studies [14,15,16].

The nucleotide-binding oligomerization domain-like receptor (NLR) family of proteins (NLRPs) is involved in the regulation of innate immunity responses. Inflammasomes are composed of the NLRP sensor, the signaling adapter apoptosis associated speck-like protein containing a caspase recruitment domain (PYD and CARD domain containing, ASC), and the caspase 1 protease [17]. Caspase 1 is activated within the inflammasome multiprotein complex through interaction with ASC. Activation of caspase 1 leads to the processing interleukin (IL)-1β precursor (pro-IL-1β) and maturation of IL-1β [17]. Increased reactive oxygen species (ROS) promote α-synuclein to form fibrils or aggregates that can be up-taken by microglia to activate the microglial NLR family pyrin domain containing 3 (NLRP3) inflammasome, which further contributes to the generation of mitochondrial ROS [18]. While reactive microglia are the major source of NLRP3 inflammasome, NLR family pyrin domain containing 1 (NLRP1) is mainly generated by neurons. It has been shown that hyperglycemia inducing neuroinflammation through activation of NLRP1 causes diabetes-associated neuron injury [19].

IL-1β-mediated transduction pathways include NF-κB (P65/P50 heterodimer), c-Jun N-terminal kinase (JNK)/proto-oncogene c-Jun (JUN), and mitogen-activated protein kinase 14 (P38)/signal transducer and activator of transcription 1 (STAT1) signaling [20]. When the NF-κB inhibitor alpha (IκBα) is phosphorylated by IκB kinase (IKK), its degradation is promoted, which permits dissociation of IκBα with NF-κB and translocation of NF-κB into nucleus with subsequent transcription of the downstream pro-inflammatory cytokines and chemokines, such as tumor necrosis factor (TNF)-α, IL-6, and C-C motif chemokine ligand 2 (MCP-1), all of which have a central role in immune response and inflammation-associated diseases [21]. Stress-activated JNK and P38 mitogen-activated protein kinase (MAPK) play an important role in inflammation-responses including further inflammatory genes transcription and cytokine production [22,23]. IL-6 binds to IL-6 receptor to promote phosphorylation of Janus kinase 2 (JAK2) (*p*-JAK2) and signal transducer and activator of transcription 3 (STAT3) (*p*-STAT3) that is inhibited by suppressor of cytokine signaling 3 (SOCS3), and *p*-STAT3 promotes expression of downstream pro-inflammatory genes [24]. 

We proposed that α-synuclein can be up-taken by microglia to activate microglia, which will then release IL-1β, IL-6, and TNF-α and can activate NLRP1 and NLRP3 in PD cellular models, both of which contribute to neuronal cytotoxicity. Previously, we have discovered Chinese herbal medicines (CHMs) Shaoyao Gancao Tang (SG-Tang) and *Glycyrrhiza inflata* (*G. inflata*), as well as pure compounds glycyrrhetic acid (C_30_H_46_O_4_) and VB-037 (C_24_H_20_N_4_O_3_) with anti-inflammatory and anti-oxidative effects. SG-Tang, a formulated CHM made of *Paeonia lactiflora* and *Glycyrrhiza uralensis* at 1:1 ratio, protects neurons from tau oligomers/aggregates-induced inflammatory damage [25] and from expanded polyglutamine-induced cytotoxicity [26]. We have also shown that extract of *G. inflata* inhibits aggregation by upregulating PPARGC1A and NFE2L2–ARE pathways in cell models of spinocerebellar ataxia 3 [27], and reduces Aβ misfolding and provides neuroprotection through anti-oxidative and anti-inflammatory action in cell models of Alzheimer’s disease [28]. Glycyrrhetic acid is a hydrolytic product of glycyrrhizic acid. As an active constituent of *G. inflata* [27] and SG-Tang [25], glycyrrhizic acid has been used to treat inflammatory diseases [29]. VB-037 is a quinoline compound protecting neurons against Aβ aggregates-induced cytotoxicity through reduction of P38- and JNK-mediated inflammation [30]. In addition, anandamide transport inhibitor AM404 (C_26_H_37_NO_2_) displays anti-inflammatory and neuroprotection effects on N-methyl-D-aspartic acid (NMDA)-induced excitotoxicity [31]. In the present study, we examined the anti-inflammatory effects of AM404, VB-037, glycyrrhizic acid, SG-Tang, and *G. inflata* on BV-2 microglia and inducible A53T SNCA-GFP-expressing SH-SY5Y cells. We also explored if these CHMs or compounds exert their effects through mitigating the NLRP1/3, caspase 1, IL-1β, IL-6, and associated IκBα/P65, JNK/JUN, P38/STAT1, and JAK2/STAT3 pathways.

## 2. Results

### 2.1. α-Synuclein Induced Microglial Activation in Mouse BV-2 Cells

α-Synuclein has been known to induce neuroinflammation and cell death [32]. In the brain, activated microglia release proinflammatory mediators in response to neuroinflammation [33]. To investigate α-synuclein induced inflammation in mouse BV-2 microglial cells, α-synuclein fibrils prepared from the *Escherichia coli*-derived α-synuclein [34] were used to activate BV-2 cells (Figure 1a). As shown in Figure 1b, the resting BV-2 microglia exhibited a ramified morphology. After lipopolysaccharide (LPS) (1 μg/mL; as a positive control) or α-synuclein fibrils (2.5–5 µM) stimulation for 20 h, cells were activated and became elongated with extended processes. Significantly increased production of nitric oxide (NO) in the cultured medium (from 3.4 µM to 9.3–11.1 µM, *p* = 0.006−0.002) and increased expression of the microglia marker Iba1 (192–219%, *p* = 0.041−0.030) were observed after LPS or α-synuclein stimulation (Figure 1c,d). Exposure of BV-2 cells to α-synuclein (2.5 µM) also resulted in increased expression of CD68 (252%, *p* = 0.004) and major histcompatibility complex class II (MHCII) (198%, *p* = 0.002) (Figure 1e). α-Synuclein fibrils at 2.5 µM was selected to provide inflammatory stimulus to BV-2 microglia for the following experiments.

### 2.2. Anti-Inflammatory Potentials of Test Compounds and Herbs in BV-2 Microglia

Three compounds (AM404, VB-037 and glycyrrhetic acid; Figure 2a) and two herbs (*G. inflata* extract and formulated SG-Tang) were tested. Ammonium glycyrrhizinate, a common active constituent in both *G. inflata* and SG-Tang, is a glycyrrhizic acid salt. It is hydrolyzed by intestinal flora to glycyrrhetic acid. The amounts of ammonium glycyrrhizinate in these two herbs were 2.23% (26.6 mM) in *G. inflata* extract [27] and 2.43% (14.52 mM) in formulated SG-Tang [25]. As shown in Figure 2b, AM404, VB-037, glycyrrhetic acid, *G. inflata,* and SG-Tang had half maximal inhibitory concentration (IC_50_) for BV-2 viability of >100 µM, 71 µM, >100 µM, >500 µg/mL and >500 µg/mL, respectively, in BV-2 cells. The anti-inflammatory effects of these compounds and herbs on α-synuclein fibrils (2.5 µM)-activated BV-2 microglial cells were examined (Figure 2c). Treatment with VB-037 (10 µM), *G. inflata* (500 µg/mL) and SG-Tang (500 µg/mL) significantly reduced NO production (from 8.7 µM to 5.0–4.2 µM, *p* = 0.018−0.002) (Figure 2d), whereas treatment with VB-037 (10 µM), glycyrrhetic acid (10 µM), *G. inflata* (500 µg/mL) and SG-Tang (500 µg/mL) significantly reduced IL-1β maturation (from 100% to 65–49%, *p* = 0.002 − <0.001) (Figure 2e).

### 2.3. Cytokine Expression Profiles in α-Synuclein-Stimulated BV-2 Cells

Mouse cytokine antibody arrays were used to examine the effect of VB-037/glycyrrhetic acid treatment on the expression of cytokine gene changes in BV-2 cells stimulated by α-synuclein fibrils. Of 40 cytokines examined, IL-1α, IL-1β, TNF-α, interferon (IFN)-γ, granulocyte-macrophage colony-stimulating factor (GM-CSF), IL-6 and granulocyte colony-stimulating factor (G-CSF) showed 1.7–104.9-fold changes in their expression levels upon addition of α-synuclein fibrils (2.5 µM), and these fold changes were decreased with VB-037 or glycyrrhetic acid (10 µM) treatment (Table 1). The protein levels of IL-1α, IL-1β, TNF-α and IFN-γ with and without compound and/or α-synuclein fibrils treatment were then examined using Western blot. VB-037/glycyrrhetic acid (10 µM) treatment significantly reduced IL-1α (62–37%, *p* < 0.001), IL-1β (58–48%, *p* = 0.001 − <0.001), TNF-α (69–51%, *p* = 0.003 − <0.001) and IFN-γ (74–71%, *p* = 0.001 − <0.001) protein levels (Figure 3a).

The GM-CSF, IL-6 and G-CSF mRNA and protein expression levels in BV-2 cells were examined using qRT-PCR and ELISA. As shown in Figure 3b, addition of VB-037 or glycyrrhetic acid (10 µM) significantly attenuated GM-CSF (25–20%, *p* < 0.001), IL-6 (65–38%, *p* = 0.005 − <0.001) and G-CSF (57–36%, *p* = 0.003 − <0.001) mRNA levels in BV-2 cells. Accordingly, GM-CSF (40–31%, *p* < 0.001), IL-6 (68–46%, *p* = 0.002 − <0.001) and G-CSF (83–68%, *p* = 0.046–0.003) protein levels in BV-2 cells were reduced with the addition of VB-037 or glycyrrhetic acid.

The release of IL-1β, TNF-α, GM-CSF, IL-6 and G-CSF cytokines from cell to culture medium was also examined using ELISA. The addition of VB-037 or glycyrrhetic acid (10 µM) significantly mitigated the release of IL-1β (75–65%, *p* = 0.001 − <0.001), TNF-α (68–47%, *p* < 0.001), GM-CSF (47–38%, *p* = 0.001 − <0.001), IL-6 (79–52%, *p* = 0.039 − <0.001) and G-CSF (90–50%, *p* = 0.012 − <0.001) from cell to culture medium (Figure 3c).

### 2.4. SH-SY5Y Cells with Induced A53T α-Synuclein-GFP Expression

Lentivirus containing in-frame A53T SNCA-GFP (Figure 4a) was used to transduce human neuroblastoma SH-SY5Y cells. The expanded blasticidin-resistance clones were examined for A53T SNCA-GFP expression after doxycycline (10 µg/mL) induction for 2 days (Figure 4b). Among them, clone 8 was further examined for dopaminergic (DAergic) neuronal marker tyrosine hydroxylase (TH) expression with 120 nM 12-O-tetradecanoylphorbol-13-acetate (TPA, also called phorbol 12-myristate 13-acetate) treatment (Figure 4c). On day 14, increased expression of TH was observed (Figure 4d), with a fifteen-fold increase of TH-fluorescence by immunostain (Figure 4e).

### 2.5. Reduction of α-Synuclein Aggregation of the Test Compounds/Herbs in A53T SNCA-GFP SH-SY5Y Cells

The aggregation-reducing potential of the test compounds/herbs was evaluated using the established A53T SNCA-GFP-expressing SH-SY5Y cells. On day 8, cells were treated with VB-037, glycyrrhetic acid (10 µM), *G. inflata* or SG-Tang (500 µg/mL) for 8 h, followed by inducing A53T SNCA-GFP expression with doxycycline (10 µg/mL) and seeding A53T SNCA-GFP aggregates with preformed α-synuclein fibrils (0.1 µM). Fluorescent images were automatically recorded by high content analysis (HCA). In addition, α-synuclein aggregates were assessed by filter trap assay (Figure 5a). Upon stain with ProteoStat, a dye better suited to smaller aggregates [35], induced A53T SNCA-GFP expression significantly provoked aggregation and percentage of aggregated cells significantly increased with α-synuclein fibrils addition compared with no addition (from 9% to 26%, *p* < 0.001), and application of test compound or herb led to 7–16% (*p* = 0.017 − <0.001) reduction of aggregated cells in A53T SNCA-GFP-expressing SH-SY5Y cells (Figure 5b,c). When protein samples from these cells were subjected to filter trap assay using GFP antibody, A53T α-synuclein-containing insoluble aggregates were also evidently reduced in cell lysates treated with test compounds or herbs (67–58%, *p* = 0.016−0.002) (Figure 5d).

### 2.6. Promotion of Neurite Outgrowth and Neuronal Survival of the Test Compounds/Herbs in A53T SNCA-GFP SH-SY5Y Cells

α-Synuclein plays a vital role in regulating neurite outgrowth [36,37]. As shown in Figure 6a,b, induction of A53T α-synuclein expression significantly reduced neurite total length (from 26.5 µm to 22.8 µm or from 100% to 86%, *p* = 0.016) and branch (from 0.67 to 0.53 or from 100% to 79%, *p* = 0.032) in SH-SY5Y cells compared to the uninduced cells, whereas augmentation of A53T α-synuclein aggregation by preformed α-synuclein fibrils did not exaggerate the reduction (*p* > 0.05). Pretreatment of test compounds or herbs significantly increased neurite length (26.3–28.6 µm or 99–108%, *p* = 0.039 − <0.001) and branch (0.75–0.87 µm or 111–129%, *p* = 0.002 − <0.001) in these aggregated A53T SNCA-GFP SH-SY5Y cells.

In addition to neurite outgrowth, lactic dehydrogenase (LDH) release, ROS, and caspase 1/3 activities were also evaluated (Figure 6c). A53T α-synuclein overexpression increased LDH release (296%, *p* < 0.001), caspase 1 (157%, *p* = 0.001) and caspase 3 (157%, *p* = 0.017) activities. α-Synuclein fibrils addition further raised LDH release (490%, *p* < 0.001), caspase 1 (213%, *p* = 0.001) and caspase 3 (263%, *p* < 0.001) activities. Significant ROS production was also observed with A53T α-synuclein overexpression plus α-synuclein fibrils addition (142%, *p* = 0.004). Application of test compounds or herbs attenuated the LDH release (304–289%, *p* < 0.001), ROS production (112–104%, *p* = 0.044 − 0.008) and caspase 3 activity (167–107%, *p* < 0.001). Significant caspase 1 reduction was also observed with VB-037, *G. inflata* and SG-Tang treatment (158–142%, *p* = 0.001 − <0.001). Together, these results demonstrate that VB-037, glycyrrhetic acid, *G. inflata,* and SG-Tang could reduce α-synuclein aggregation, increase neurite outgrowth, and protect cells from cell death in aggregated A53T SNCA-GFP SH-SY5Y cells.

### 2.7. Downregulation of NLRP1/3 Inflammasome Pathways by Test Compounds/Herbs in A53T SNCA-GFP SH-SY5Y Cells

Both NLRP1 and NLRP3 inflammasomes promote the activation of caspase 1 and the subsequent proteolytic processing and release of IL-1β [38]. Release of the mature form of IL-1β, via NLRP3 inflammasome activation, can be induced in monocytes by oxidative stress and fibrillar α-synuclein [39]. While NLRP3 is mainly expresses in activated microglia, NLRP1 has been shown to be activated in neurons by Aβ aggregates to cleave caspase 1 into its active form, which would drive IL-1β maturation and subsequent neuroinflammation in AD [40]. We thus examined the expression of NLRP1, NLRP3, ASC, and IL-1β proteins in α-synuclein-triggered neuroinflammation of A53T SNCA-GFP SH-SY5Y cells. As shown in Figure 7a, induced expression of A53T α-synuclein in SH-SY5Y cells increased the expressions of NLRP1 (197%, *p* = 0.002) and IL-1β (148%, *p* = 0.047). These up-regulations were further exaggerated by α-synuclein fibrils addition (NLRP1: 282%, *p* = 0.003; IL-1β: 201%, *p* = 0.022), whereas treatment with test compounds or herbs attenuated the levels of NLRP1 (194–160%, *p* = 0.004 − <0.001) and IL-1β (137–117%, *p* = 0.006 − <0.001). For NLRP3 and ASC, significant increase was only observed with induced α-synuclein expression plus α-synuclein fibrils addition (NLRP3: 144%, *p* = 0.002; ASC: 125%, *p* = 0.005). Application of test compounds or herbs attenuated the level of NLRP3 (98–82%, *p* = 0.002 − <0.01) and ASC (86–74%, *p* = 0.021–0.004).

Upon binding to the IL-1 receptor and accessory proteins, IL-1β triggers activation of P38 and JNK MAPK pathways [20], both of which play a critical role in inflammatory cell signaling [41,42]. In addition, IL-1 itself is a strong inducer of NF-κB activity [43]. Therefore, we examined the expression of these signaling pathways in α-synuclein-triggered neuroinflammation of A53T SNCA-GFP SH-SY5Y cells by immunoblotting using specific antibodies. As shown in Figure 7b, *p*-JNK (181%, *p* < 0.001), *p*-JUN (218%, *p* < 0.001), *p*-IκBα (156%, *p* = 0.006), *p*-P65 (223%, *p* < 0.001), *p*-P38 (205%, *p* < 0.001) and *p*-STAT1 (148%, *p* = 0.003) were significantly increased with induced α-synuclein expression plus α-synuclein fibrils addition, whereas treatment with test compounds or herbs reduced the *p*-JNK (from 181% to 119–105%, *p* = 0.006–0.001), *p*-JUN (from 218% to 157–118%, *p* = 0.003 − <0.001), *p*-IκBα (from 156% to 97–86%, *p* = 0.004 − <0.001), *p*-P65 (from 223% to 141–121%, *p* = 0.002 − <0.001), *p*-P38 (from 205% to 140–131%, *p* = 0.029–0.013) and *p*-STAT1 (from 148% to 114–98%, *p* = 0.037–0.002) levels. The changes in total JNK (87–112%), JUN (100–129%), IκBα (97–118%), P65 (100–126%), P38 (100–125%), and STAT1 (100–144%) were not significant (*p* > 0.05).

Dysregulated continual synthesis of IL-6 plays a pathological effect on chronic inflammation [44]. The IL-6 may signal via the JAK2/STAT3 pathway, and SOCS3 inhibits this signal transduction pathway [24]. As shown in Figure 7c, IL-6 (100%, *p* < 0.001), *p*-JAK2 (133%, *p* < 0.001) and *p*-STAT3 (165%, *p* < 0.001) were significantly increased with induced α-synuclein expression plus α-synuclein fibrils addition, whereas treatment with test compounds or herbs reduced the IL-6 (from 100% to 81–60%, *p* = 0.020 − <0.001), *p*-JAK2 (from 133% to 101–91%, *p* = 0.001 − <0.001) and *p*-STAT3 (from 165% to 107–88%, *p* = 0.001 − <0.001). The changes in total JAK2 (100–118%) and STAT3 (98–125%) were not significant (*p* > 0.05). Consistently, the reduced SOCS3 (55%, *p* < 0.001) was rescued (from 55% to 91–100%, *p* = 0.004 − <0.001) after treatment with test compounds or herbs. These results demonstrated the anti-inflammatory effects of the test compounds and herbs on PD neuronal cells.

## 3. Discussion

Several lines of evidence have shown neuroinflammation contributes to neurodegeneration in PD [45]. However, which and how inflammatory pathways are involved in the pathogenesis remain to be explored. In this study, we provided evidence that BV-2 cells are activated by α-synuclein fibrils to secrete NO and promote maturation of IL-1β. IL-1β maturation is inhibited by VB-037, glycyrrhetic acid, *G. inflata,* and SG-Tang. Results of mouse cytokine antibody arrays demonstrate that expression of IL-1α, IL-1β, TNF-α, IFN-γ, IL-6, GM-CSF and G-CSF are increased in α-synuclein-stimulated BV-2 cells, which can be attenuated by VB-037 or glycyrrhetic acid. The release of IL-1β, TNF-α, GM-CSF, IL-6 and G-CSF cytokines from BV-2 cells to culture medium is also reduced by VB-037 or glycyrrhetic acid. It is noted that low IC_50_ cytotoxicity of VB-037, glycyrrhetic acid, *G. inflata,* and SG-Tang in cells indicate their potential as agents for treatment of neurodegenerative diseases including PD.

We then showed that α-synuclein fibrils could provoke aggregation and increase oxidative stress, leading to impaired neurite outgrowth and apoptosis by activating NLRP1/NLRP3 inflammasome, IL-1β-mediated IκBα/P65, JNK/JUN, P38/STAT1, and IL-6-mediated JAK2/STAT3 pathways in DAergic neurons derived from α-synuclein-expressing SH-SY5Y cells. Furthermore, the aggregation, oxidative stress, neurite outgrowth deficits, apoptosis and neuroinflammation can be ameliorated by treatment with the tested compounds or CHM. VB-037 has been shown to inhibit LPS/IFN-γ-induced activation of BV-2 cells and attenuate IL-1β-, caspase 1-, P38- and JNK-mediated inflammatory damage caused by Aβ aggregates [30]. This study further confirms its anti-inflammation effect on α-synuclein-activated BV-2 and α-synuclein-expressing SH-SY5Y cells. Anti-inflammatory activity of SG-Tang and *G. inflata* has also been shown in cell models induced by tau or Aβ misfolding [25,28]. Here, we demonstrated further evidence that *G. inflata* and SG-Tang provide neuroprotection through anti-inflammatory activity in PD cellular models. Glycyrrhetinic acid (glycyrrhetic acid) has been shown to act like a dopamine receptor D3 agonist to restore dopaminergic function [46]. Although glycyrrhizic acid reduced IL-1β, IL-6, and TNF-α, and the number of activated astrocytes and microglia in rotenone-injected animals [47], whether its metabolite glycyrrhetic acid also has anti-inflammation effects on PD models is not known. Here we showed that glycyrrhetic acid exhibits anti-inflammatory action to mitigate neurotoxicity induced by α-synuclein fibrils in both BV-2 and A53T SNCA-GFP SH-SY5Y cells.

While NLRP3 is mainly produced by microglia and its role in contributing to neurodegeneration has been well shown in PD models [18,39], NLRP1 is majorly activated in neurons by different toxic stimuli and its involvement in neuroinflammation is less investigated [48,49]. NLRP1 receptor is increased in human AD brains and neurons, and NLRP1 up-regulation to activate caspase 1 to cleave pro-IL-1β has been shown in serum-deprived neurons [50]. Aβ oligomers have been reported to induce ATP leakage in cells and cause an overexpression of P2 × 7 purinergic receptors, leading to NLRP1 activation in both of microglia and hippocampal neurons [51,52]. Aβ aggregates induce neuronal NLRP1 inflammasome that drives IL-1β maturation and subsequent neuroinflammation in an AD mouse model [40]. In accordance with the result of NLRP1 upregulated by Aβ oligomers in neurons, we found that α-synuclein fibrils increase NLRP1, NLRP3, ASC, and caspase 1 that subsequently promote IL-1β maturation in α-synuclein-expressing SH-SY5Y neurons. Caspase 1 can be activated by known inflammasome-stimulators to directly cleave α-synuclein to generate truncated species that are prone to form aggregates in a neuronal PD cell model, and silencing of caspase 1 expression rescues neurotoxicity caused by α-synuclein, suggesting that under certain toxic stimuli, inflammasome, and caspase 1 cause PD pathology in neurons [53]. Substantial evidence has shown that ROS activate NLRP3 inflammasome through release of the ROS-sensitive NLRP3 ligand thioredoxin-interacting protein (TXNIP) from its inhibitor thioredoxin and antioxidants can attenuate NLRP3-mediated inflammatory cytotoxicity [54,55,56]. We therefore suggest that α-synuclein fibrils may act like an inflammasome-stimulator or produce ROS to activate NLRP1, ASC, caspase 1, and IL-1β, leading to increased α-synuclein aggregates and neurotoxicity, all of which can be mitigated by the tested compounds and CHM as shown in the present study. However, future studies are warranted to provide direct evidence of the role of α-synuclein fibrils as an inflammasome-stimulator.

Previously, studies have shown sustained IL-1β expression at pro-inflammatory level confers profound toxic effects on the substantia nigra in two different animal models [57,58]. As shown above in our study results, mature IL-1β can be released by activated BV-2 cells stimulated by α-synuclein fibrils, or produced by neurons probably through NLRP1-caspase 1 pathway. Although increased IL-1β has been shown in the brains of human PD and mouse PD models [59,60,61], whether the downstream signaling pathways of IL-1β in neurons overexpressing α-synuclein are involved is not clear. Responding to IL-1β binding signal, JNK, or P38 is activated and translocates to the nucleus to phosphorylate transcription factors such as JUN, FOS, STAT1, and MYC, which subsequently up-regulate the expression of pro-apoptotic genes [23,62]. Cytokines and *p*-P38 are increased by 6-OHDA and *Toona sinensis* seeds exert anti-inflammatory effects through suppressing *p*-P38 in a PD rat model [63]. Nuclear activation of *p*-P65 has been shown to mediate inflammatory toxicity in MPTP-, LPS- and rotenone-induced PD animal models and agents provide neuroprotection via inhibiting *p*-P65 activation [64]. Downstream pathways of IL-1β including *p*-IκBα/*p*-P65, *p*-JNK/*p*-JUN and *p*-P38/*p*-STAT1 were augmented in A53T SNCA-GFP SH-SY5Y cells added with α-synuclein fibrils, which are attenuated by VB-037, glycyrrhetic acid, *G. inflata* and SG-Tang, indicating the anti-inflammatory effects of the tested compounds and CHM on A53T SNCA-GFP SH-SY5Y cells via targeting IL-1β and its downstream signaling pathways.

IL-6 has been identified as an important cytokine to coordinate the acute phase response and the activation of immunocompetent glia within the brain. IL-6 is expressed not only by glial cells but also by neurons and can be activated by IL-1β [65,66,67]. The pro-inflammatory effects of IL-6 are mediated through autoactivation of JAK2/JAK3, which phosphorylate STAT3/STAT1 [24]. STAT3 and STAT1 are latent transcription factors and once phosphorylated, they translocate into the nucleus and induce the transcription of IL-6-responsive genes, whereas SOCS3 counteracts JAK2/STAT3 signaling to diminish the IL-6-mediated inflammation [24]. IL-6-mediated JAK2/STAT3 signaling cascade has been shown to contribute to neurodegeneration in AD and Huntington’s disease models [68]. Increased IL-6 has been shown in the brains, CSF and plasma of human PD, and brains of mouse PD models [59,69,70], whether the downstream signaling pathways of IL-6 are involved, remains to be investigated. Overexpression of α-synuclein activates STAT3 in the substantia nigra in a mouse PD model, and pro-inflammatory cytokines and cytotoxicity are reduced by miR-let-7a that suppresses STAT3 [71]. Similarly, our study has shown that IL-6 and JAK2/STAT3 are induced and SOCS3 is reduced by α-synuclein fibrils, both of which can be rescued by VB-037, glycyrrhetic acid, *G. inflata,* and SG-Tang, indicating these compounds and CHM protect A53T SNCA-GFP SH-SY5Y cells against inflammatory damage via targeting IL-6 and its downstream signaling pathway. However, another study shows that bisdemethoxycurcumin protects neurons from rotenone-induced PD pathology via enhancing JAK2/STAT3 signaling [72]. These results suggest the role of IL-6/JAK2/STAT3 signaling pathway in PD pathogenesis remains to be clarified by future studies.

## 4. Materials and Methods

### 4.1. Compounds, Herbs, and Cell Culture

Anandamide transport inhibitor AM404 and pentacyclic triterpenoid glycyrrhetic acid were purchased from Sigma-Aldrich (St. Louis, MO, USA), and quinoline compound VB-037 was purchased from Enamine (Kiev, Ukraine). Aqueous extract from *G. inflata* [27] and formulated SG-Tang [25,26] was provided by Sun-Ten Pharmaceutical Company (Taipei, Taiwan). Mouse BV-2 microglial cells (kind gift from Dr. Han-Min Chen, Catholic Fu-Jen University, New Taipei City, Taiwan) were routinely maintained in Dulbecco’s modified Eagle medium (DMEM) supplemented with 10% fetal bovine serum (FBS) (Invitrogen, Carlsbad, CA, USA). Human neuroblastoma SH-SY5Y cells (ATCC CRL-2266) were maintained in DMEM/Nutrient mixture F-12 (DMEM/F-12) supplemented with 10% FBS. Cells were cultured at 37 °C with 95% relative humidity and 5% CO_2_.

### 4.2. Cell Proliferation Assay

To evaluate compound cytotoxicity, 5 × 10^3^ BV-2 cells were plated on 48-well dishes, grown for 20 h, and treated with the test compounds (1–100 μM) or herbs (5–500 μg/mL). After 1 day, 20 μL of 3-(4,5-dimethylthiazol-2-yl)-2,5-diphenyltetrazolium bromide (MTT) (5 mg/mL) was added to the cells at 37 °C for 4 h. 200 μL of lysis buffer (10% Triton X-100, 0.1 N HCl, 18% isopropanol) was then added to dishes and the absorbance of the insoluble purple formazan product at OD 570 nm was read by a FLx800 fluorescence microplate spectrophotometer (Bio-Tek, Winooski, VT, USA).

### 4.3. Detection of BV-2 Microglial Activation

To detect microglial activation, 1 × 10^5^ BV-2 cells were plated on 12-well dishes in DMEM containing 1% FBS, grown for 20 h, and treated with LPS (1 μg/mL) or α-synuclein fibrils (2.5–5 µM). After 1 day, the release of NO in the media was evaluated by Griess assay according to the manufacturer’s protocol (Sigma-Aldrich). Levels of Iba1 in cells were examined by Western blot as described below using Iba1 antibody (1:500; Wako, Osaka, Japan). In addition, BV-2 cells without or with α-synuclein (2.5 µM) treatment were fixed (4% paraformaldehyde), permeabilized (0.1% Triton X-100), blocked (2% bovine serum albumin), and stained with primary anti-CD68 (1:1000; Cell Signaling*,* Danvers, MA, USA) or anti-MHCII (1:1000; Invitrogen, Waltham, MA, USA) antibody, followed by secondary antibody conjugated to Cy5 (1:1000; Invitrogen). Nuclei were detected using DAPI (0.1 μg/mL; Sigma-Aldrich). Cells were examined using a Zeiss LSM 880 confocal laser scanning microscope (Carl Zeiss Microscopy, Oberkochen, Germany).

### 4.4. Mouse Inflammation Antibody Array

Protein samples from BV-2 cells with different treatments (no added α-synuclein, α-synuclein added, VB-037 pretreated plus α-synuclein added, glycyrrhetic acid pretreated plus α-synuclein added) were prepared and incubated with mouse inflammation antibody array membranes (RayBiotech, Norcross, GA, USA). The relative levels of 40 inflammation-related cytokines in BV-2 cell lysates were measured with the array. The detected changes were confirmed through Western blot using specific antibodies to IL-1α (1:2000; R&D Systems, Minneapolis, MN, USA), IL-1β (1:1000; Abcam), TNF-α (1:2000; Abcam), and IFN-γ (1:1000; Abcam), using β-tubulin (1:1000; Sigma-Aldrich) as a loading control. In addition, changes in GM-CSF, IL-6 and G-CSF were confirmed through real-time reverse transcription-PCR (qRT-PCR) and enzyme-linked immunosorbent assay (ELISA) as described below.

### 4.5. Cytokine qRT-PCR Assay

To measure GM-CSF, IL-6 and G-CSF RNA in BV-2 cells, total RNA was extracted using TRIzol reagent, treated with DNase to remove chromosomal DNA, and used for cDNA synthesis with SuperScript III reverse transcriptase (Thermo Fisher Scientific, Waltham, MA, USA). Relative cytokine RNA expression was analyzed in 100 ng cDNA through real-time PCR (StepOnePlus Real-time PCR system; Applied Biosystems, Foster City, CA, USA) with TaqMan fluorogenic probes Mm01290062_m1 for Csf2 (GM-CSF), Mm00446190_m1 for IL-6, Mm00438334_m1 for Csf3 (G-CSF), and Mm00607939_s1 for β-actin control (Thermo Fisher Scientific). Fold change was calculated using the formula 2^ΔCt^, ΔC_T_ = C_T_ (control) - C_T_ (target), in which C_T_ indicates cycle threshold.

### 4.6. Cytokine ELISA

The levels of GM-CSF, IL-6, and G-CSF in BV-2 cell lysates and IL-1β, TNF-α, GM-CSF, IL-6, and G-CSF in cell culture media were determined using ELISA. Specifically, mouse IL-1β Instant ELISA, mouse IL-6, TNF-α, and GM-CSF (Csf2) Platinum ELISA, and mouse G-CSF (Csf3) ELISA kits (Thermo Fisher Scientific) were used. All experimental procedures were performed following the corresponding manufacturers’ instructions. The optical density at 450 nm was detected using a microplate reader (Multiskan Go, Thermo Fisher Scientific).

### 4.7. A53T SNCA-GFP Construct

The A53T mutation is a point mutation of α-synuclein that enhances its rate of fibrillization, leading to early-onset PD [73]. A53T mutation was introduced into SNCA cDNA [34] by site-directed mutagenesis (QuikChange II XL mutagenesis kit; Stratagene, La Jolla, CA, USA) using primer 5′-GTGGTGCATGGTGTGACAACAGTGGCTGAGAAG-3′ (mutated nucleotide underlined) and sequenced. Then the A53T SNCA cDNA acid (and GFP (from pEGFP-N1, Invitrogen) were subcloned into pcDNA5/FRT/TO (Invitrogen) to generate in-frame A53T SNCA-GFP. The A53T SNCA-GFP was then recombined into the *Nhe*I and *Pme*I sites of lentiviral vector pAS4.1w.Pbsd-aOn (National RNAi Core Facility, Institute of Molecular Biology/Genomic Research Center, Academia Sinica) and used to prepare lentivirus according to the standard protocol.

### 4.8. A53T SNCA-GFP SH-SY5Y Cells

To establish human cell line with inducible A53T SNCA-GFP expression, SH-SY5Y cells were transduced with the lentivirus carrying A53T SNCA-GFP. Briefly, SH-SY5Y cells were plated into 12-well (5 × 10^4^/well) dishes, grown for 20 h, and transduced with lentivirus (0.01 multiplicity of infection) in the presence of polybrene (8 µg/mL; Sigma-Aldrich). After 6 h, the medium was changed into fresh media. Next day, the infected cells were passaged into a 10-cm dish, followed by selection of stable transfectants with blasticidin (5 μg/mL; InvivoGen, San Diego, CA, USA). Fresh blasticidin-containing medium was added every 3 to 4 days. This selection step lasted until the untransduced control cells were completely dead. Growing blasticidin-resistant clones from single cell were picked, expanded, and examined for doxycycline (10 µg/mL; Sigma-Aldrich) induced SNCA/A53T-GFP expression using anti-α-synuclein (1:1000; BD Biosciences, Franklin Lakes, NJ, USA) and anti-GFP (1:500; Santa Cruz, Santa Cruz, CA, USA) antibodies.

### 4.9. Tyrosine Hydroxylase Staining

DAergic differentiation of SH-SY5Y cells was promoted by applying TPA (120 nM; Enzo Life Sciences, Farmingdale, NY, USA) [74]. TPA-treated A53T SNCA-GFP SH-SY5Y cells were fixed (4% paraformaldehyde), permeabilized (0.1% Triton X-100), blocked (2% bovine serum albumin), and stained with TH antibody (1:1000; Millipore, Billerica, MA, USA), followed by secondary antibody conjugated to Cy5 (Invitrogen). Nuclei were counterstained with DAPI. DAergic differentiation was examined by simultaneous fluorescent imaging of GFP (482/35 nm excitation and 536/40 nm emission) and Cy5 (628/40 nm excitation and 692/40 nm emission) fluorescence using a HCA system (ImageXpress Micro; Molecular Devices, San Jose, CA, USA). Moreover, 5 × 10^4^ A53T SNCA-GFP expressing cells (green) from three independent experiments were analyzed for TH expression (red).

### 4.10. α-Synuclein Aggregation and Neurite Outgrowth Analyses

A53T SNCA-GFP SH-SY5Y cells were seeded in 12-well (5 × 10^4^/well) plate, with TPA (120 nM) added next day. On day 8, cells were treated with VB-037, glycyrrhetic acid (10 µM), *G. inflata,* or SG-Tang (500 µg/mL) for 8 h, followed by doxycycline (10 µg/mL) addition to induce A53T SNCA-GFP expression. In addition, 0.1 µM preformed α-synuclein fibrils were added to seed the formation of A53T SNCA-GFP aggregates in the cells [75]. The cells were kept in the medium containing TPA, doxycycline and compound or herb for 6 days. Then cells were fixed and permeated as described, and stained with ProteoStat dye (1/5000; Enzo Life Sciences). After detecting nuclei with DAPI, percentages of aggregated cells were assessed using the ImageXpress Micro HCA system, with 482/35 nm excitation and 536/40 nm emission for GFP, and 543/22 nm excitation and 593/40 nm emission for ProteoStat. In addition, compound/herb-treated SH-SY5Y cells were collected and α-synuclein aggregates in cell lysates were examined by filter trap assay using GFP antibody (1:500; Santa Cruz).

For neurite outgrowth analysis, the fixed and permeated cells were stained with TUBB3 antibody (1:1000; Covance, Princeton, NJ, USA), followed by anti-rabbit Alexa Fluor ^®^555 antibody (1:1000; Thermo Fisher Scientific). After nuclei staining, neuronal images were captured and analyzed (Neurite Outgrowth Application Module; Molecular Devices).

### 4.11. Caspase 1 and 3 Activities and LDH Release Assays

TPA-differentiated A53T SNCA-GFP SH-SY5Y cells (4 × 10^5^ on 6-well dishes for caspase 1 activity and LDH release assays; 5 × 10^4^ on 12-well dishes for caspase 3 activity assay) were pretreated with test compounds or herbs and A53T SNCA-GFP expression was induced in the presence of α-synuclein fibrils as described. For LDH release assay, cell culture media were collected on day 14 and the release of LDH was examined by using LDH cytotoxicity assay kit (Cayman, Ann Arbor, MI, USA). The absorbance was read at 490 nm with Multiskan GO microplate reader. For caspase 1 and 3 activity assays, cells were lysed by repeated cycles of freezing and thawing. Caspase 1/3 activities were measured with the caspase 1 (BioVision, Milpitas, CA, USA) and caspase 3 (Sigma-Aldrich) fluorimetric assay kits, with 420/50 nm excitation and 485/20 nm emission (caspase 1 assay) or 360/40 nm excitation and 460/40 nm emission (caspase 3 assay) (FLx800 fluorescence microplate reader, Bio-Tek).

### 4.12. ROS Analysis

To measure ROS activity in cells, the A53T SNCA-GFP SH-SY5Y cells (5 × 10^4^ on 12-well dishes) were incubated at 37 °C for 30 min in the fluorogenic CellROX Deep Red Reagent (5 µM; Molecular Probes, Eugene, OR, USA). After washing with PBS, cells were analyzed for red (ROS) fluorescence on a flow cytometry system (Becton-Dickinson, Franklin Lakes, NJ, USA), with 633 nm excitation and 661/16 nm emission. Each sample contained 2 × 10^4^ cells.

### 4.13. Western Blot Analysis for Inflammasome Signaling

Cells were lysed using buffer (50 mM Tris-HCl pH 8.0, 150 mM NaCl, 1 mM EDTA pH 8.0, 0.1% SDS, 0.5% sodium deoxycholate, 1% Triton X-100) containing the protease inhibitor cocktail (Sigma-Aldrich). After sonication, the lysates were centrifuged at 12,000× *g* for 10 min at 4 °C and protein concentrations determined (protein assay kit; Bio-Rad, Hercules, CA, USA). Total proteins (20 µg) were electrophoresed on 10% SDS-polyacrylamide gel and transferred onto polyvinylidene difluoride membrane (Sigma-Aldrich) by reverse electrophoresis. After being blocked, the membrane was stained with NLRP1 (1:1000; Novus Biologicals, Centennial, CO, USA), NLRP3 (1:1000; Cell Signaling, Danvers, MA, USA), ASC (1:1000; Abcam), IL-1β (1:1000; Abcam), IL-6 (1:1000; Abcam), IκBα (1:1000; Cell Signaling), *p*-IκBα (S32/36) (1:1000; Cell Signaling), P65 (1:1000; Cell Signaling), *p*-P65 (S536) (1:1000; Cell Signaling), JNK (1:1000; Cell Signaling), *p*-JNK (T183/Y185) (1:500; Cell Signaling), JUN (1:1000; Cell Signaling), *p*-JUN (S63) (1:1000; Cell Signaling), P38 (1:1000; Cell Signaling), *p*-P38 (T180/Y182) (1:1000; Cell Signaling), STAT1 (1:1000; Cell Signaling), *p*-STAT1 (S727) (1:1000; Cell Signaling), JAK2 (1:1000; Cell Signaling), *p*-JAK2 (Y1007/1008) (1:1000; Invitrogen), STAT3 (1:500; Santa Cruz), *p*-STAT3 (Y705) (1:500; Santa Cruz), SOCS3 (1:500; Santa Cruz) or glyceraldehyde-3-phosphate dehydrogenase (GAPDH) (1:5000; MDBio, Taipei, Taiwan) antibody at room temperature 2 h or 4 °C overnight. The immune complexes were detected using goat anti-mouse or goat anti-rabbit IgG-HRP antibody (1:5000; GeneTex, Irvine, CA, USA) and chemiluminescent substrate (Millipore).

### 4.14. Statistical Analysis

For each data set, the experiments are performed three times and data were expressed as the means ± standard deviation (SD). Differences between groups were evaluated by Student’s t test (comparing two groups) or one-way analysis of variance (ANOVA) with a post hoc Tukey test where appropriate (comparing several groups). All *p* values were two-tailed, with values lower than 0.05 to be considered statistically significant.

## 5. Conclusions

In summary, our study shows that in addition to microglia activation and pro-inflammatory cytokine release, α-synuclein fibrils induce neuronal inflammation that contributes to increased aggregation, oxidative stress, and neurotoxicity in A53T SNCA-GFP SH-SY5Y cells. VB-037, glycyrrhetic acid, *G. inflata,* and SG-Tang reduce aggregation, neuroinflammation, ROS, and apoptosis, and promote neurite outgrowth by downregulating NLRP1/NLRP3-, IL-1β-, and IL-6-mediated pathways and their downstream caspase 1, IκBα/P65, JNK/JUN, P38/STAT1, JAK2/STAT3 signaling (Figure 8). The study results provide insight into the involvement of neuroinflammation in PD pathogenesis and the potential of VB-037, glycyrrhetic acid, *G. inflata,* and SG-Tang in treating PD.

## Figures and Tables

**Figure 1 ijms-22-01062-f001:**
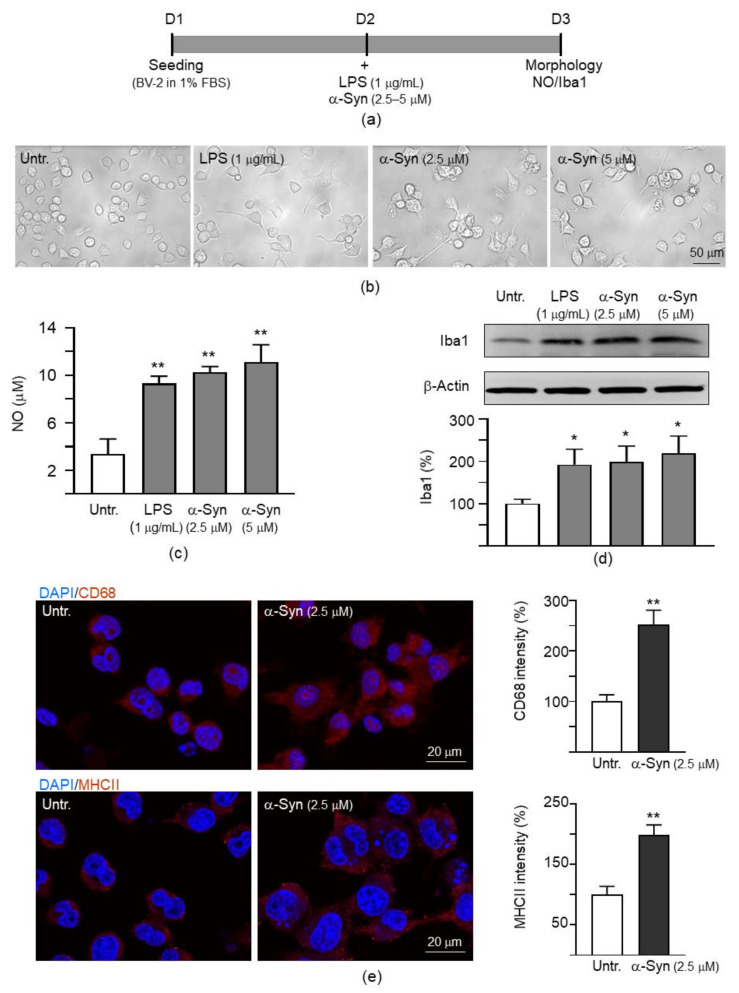
α-Synuclein induced inflammation in mouse BV-2 microglial cells. (**a**) Experimental flow chart. BV-2 cells were plated in 1% fetal bovine serum (FBS) on day 1. On day 2, the cells were treated with lipopolysaccharide (LPS) (1 μg/mL) or α-synuclein fibrils (α-Syn, 2.5–5 μM) to induce inflammation. On day 3, the cells were examined for microglial activation by morphology, NO release in cell culture medium and Iba1 Western blotting. (**b**) Morphology, (**c**) NO production (*n* = 3), and (**d**) Iba1 level (*n* = 3) of inactive and LPS or α-synuclein fibrils activated BV-2 cells. For normalization, the relative Iba1 level in inactive cells was set as 100%. (**e**) Immunocytochemistry of CD68 and major histcompatibility complex class II (MHCII) (red) and fluorescent intensity with or without α-synuclein (2.5 μM) activation. Nuclei were counter stained with 4′,6-diamidino-2-phenylindole (DAPI) (blue). For normalization, the relative fluorescent intensity in inactive cells was set as 100% (*n* = 3). *p* values: comparisons between inactive and activated cells (*: *p* < 0.05 and **: *p* < 0.01). (Two-tailed Student’s *t* test). Original blot images can be found in the Appendix A.

**Figure 2 ijms-22-01062-f002:**
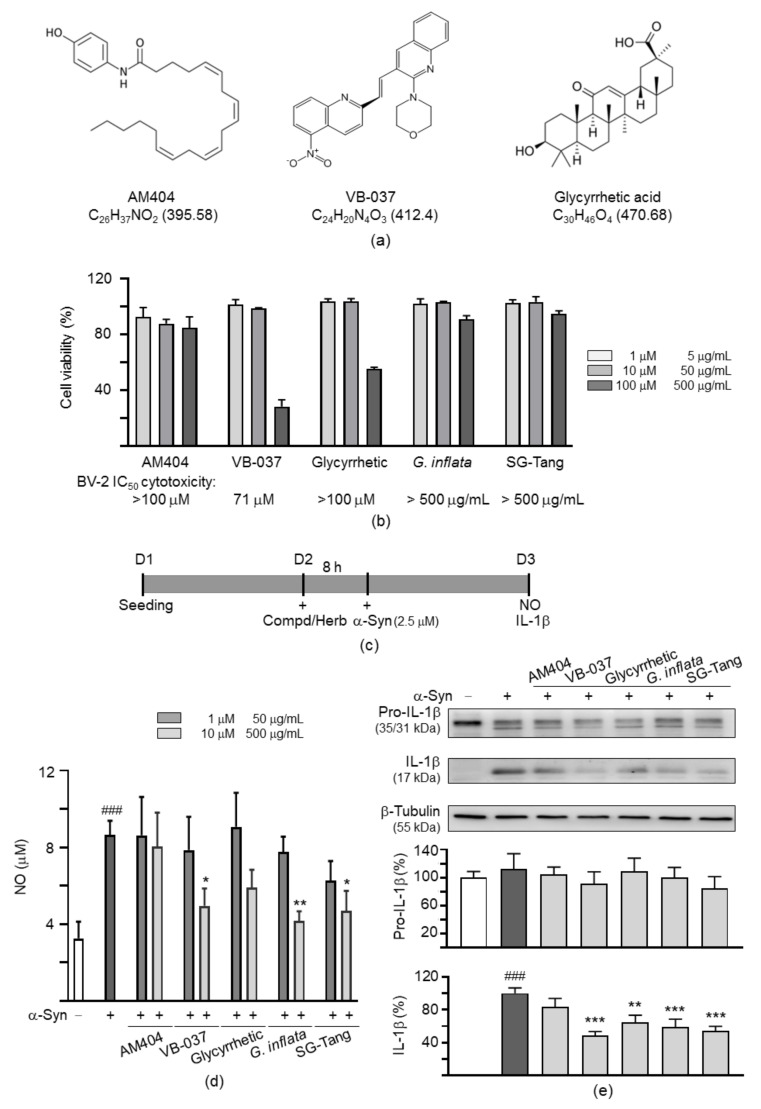
Anti-inflammatory activity of test compounds and herbs in α-synuclein fibrils activated BV-2 cells. (**a**) Structure, formula and molecular weight of test compounds AM404, VB-037 and glycyrrhetic acid. (**b**) Cytotoxicity of test compounds and herbs (*G. inflata* and SG-Tang) against BV-2 examined using the MTT assay. Cells were treated with test compound (1–100 μM) or herb (5–500 μg/mL) and cell viability was measured the next day (*n* = 3). To normalize, the relative viability of untreated cells was set at 100%. Half maximal inhibitory concentration (IC_50_) values are presented below. (**c**) Experimental flow chart. BV-2 cells were plated in 1% FBS on day 1. After 20 h, the cells were pretreated with test compound or herb for 8 h, followed by α-synuclein fibrils (α-Syn, 2.5 μM) treatment. After 20 h, the cells were examined for nitric oxide (NO) release and interleukin (IL)-1β maturation. (**d**) NO production of α-Syn-activated BV-2 cells pretreated with test compound (1–10 μM) or herb (50–500 μg/mL) (*n* = 3). For normalization, the relative NO level in α-Syn-activated cell culture medium was set as 100%. (**e**) Pro-IL-1β and IL-1β levels of α-Syn-activated BV-2 cells pretreated with test compound (10 μM) or herb (500 μg/mL) (*n* = 3). β-Tubulin was used as a loading control. For normalization, the relative pro-IL-1β and IL-1β levels in α-Syn-activated cells were set as 100%. *p* values: comparisons between with and without fibril addition (^###^: *p* < 0.001), or between with and without compound/herb treatment (*: *p* < 0.05, **: *p* < 0.01 and ***: *p* < 0.001). (One-way ANOVA with a post hoc Tukey test).

**Figure 3 ijms-22-01062-f003:**
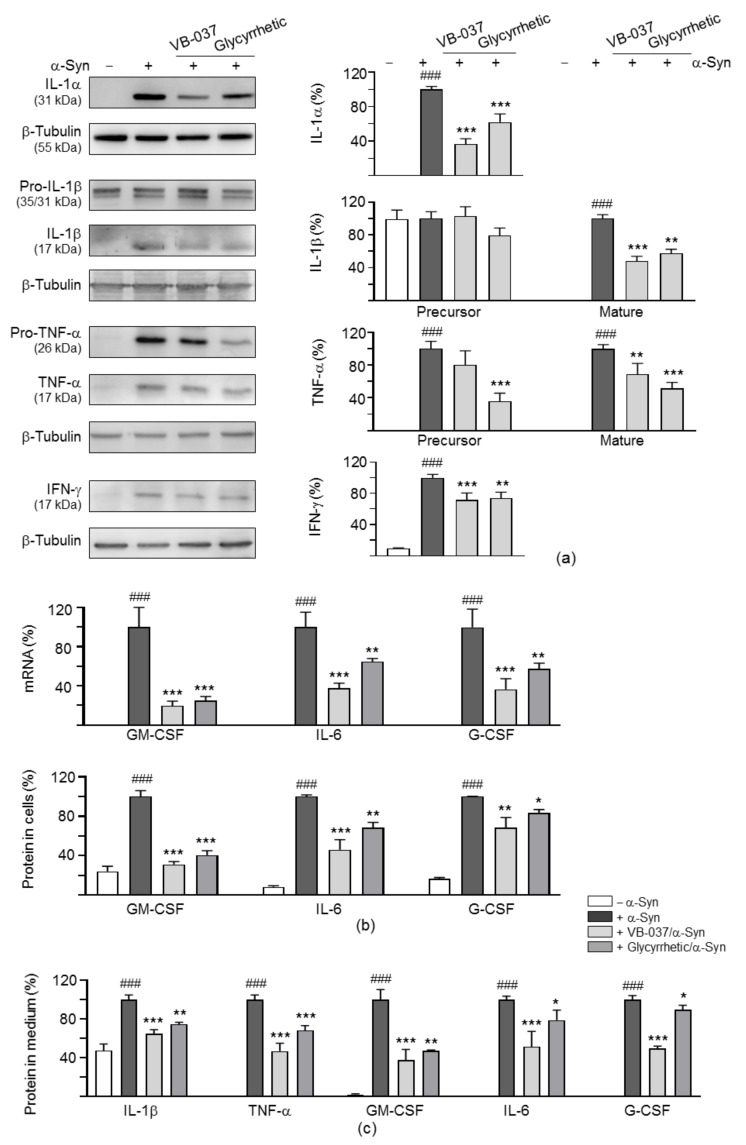
Targets of VB-037 and glycyrrhetic acid in α-synuclein-activated BV-2 cells. (**a**) Western blot analysis of interleukin (IL)-1α, IL-1β (pro- and mature forms), tumor necrosis factor (TNF)-α (pro- and mature forms), and interferon (IFN)-γ (*n* = 3). β-Tubulin was used as a loading control. (**b**) qRT-PCR (for mRNA) and ELISA (for protein) examination of granulocyte-macrophage colony-stimulating factor (GM-CSF), IL-6 and granulocyte colony-stimulating factor (G-CSF) expression (*n* = 3). (**c**) ELISA examination of IL-1β, TNF-α, GM-CSF, IL-6 and G-CSF cytokines release to cell culture medium (*n* = 3). For normalization, the relative cytokine level in α-Syn-activated cells was set as 100%. *p* values: comparisons between with and without fibril addition (^###^: *p* < 0.001), or between with and without compound/herb treatment (*: *p* < 0.05, **: *p* < 0.01 and ***: *p* < 0.001). (One-way ANOVA with a post hoc Tukey test).

**Figure 4 ijms-22-01062-f004:**
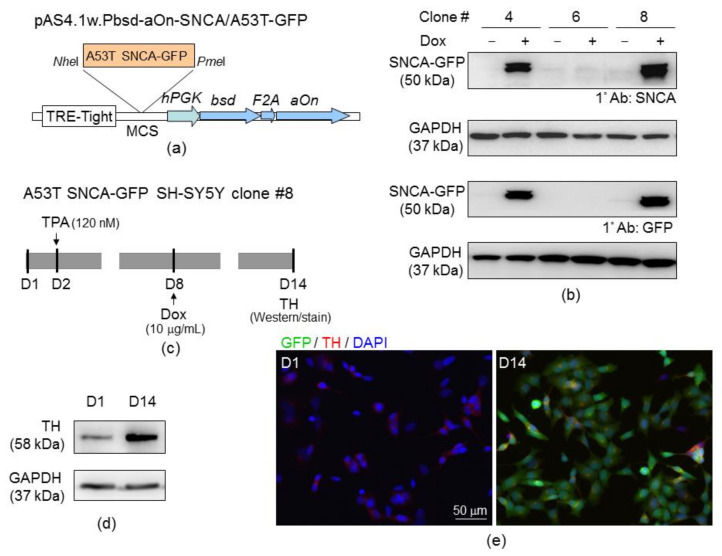
A53T SNCA-GFP SH-SY5Y cells. (**a**) Lentiviral vector with A53T SNCA-GFP cloned between *Nhe*I and *Pme*I sites in MCS (multiple cloning site) and driven by a tetracycline inducible system (TRE-Tight promoter containing 7 copies of modified tetO sequence, a tetracycline repressor binding sequence). The in-frame fused *bsd* (blasticidin, selective marker)-*aOn* (transcription factor, activating TRE-Tight in the presence of tetracycline) is under human phosphoglycerate kinase (*hPGK*) promoter. F2A protease cleaves fusion protein into functional bsd and aOn. (**b**) Western blot images of A53T SNCA-GFP SH-SY5Y cell clones 4, 6 and 8 using α-synuclein (SNCA) and GFP antibodies after induction of expression for two days (+ Dox, 10 µg/mL). Glyceraldehyde 3-phosphate dehydrogenase (GAPDH) was used as a loading control. **(c**) Experimental flow chart for DAergic differentiation. On day 2 (D2), neuronal differentiation was promoted with 12-O-tetradecanoylphorbol-13-acetate (TPA) (120 nM) for 13 days. On day 8 (D8), A53T SNCA-GFP expression was induced with doxycycline (10 µg/mL) for 6 days. On day 14 (D14), tyrosine hydroxylase (TH) expression was examined. (**d**) TH Western on days 1 and 14 using GAPDH as a loading control. (**e**) TH stain (red) on days 1 (D1) and 14 (D14) in A53T SNCA-GFP SH-SY5Y cells. Nuclei were detected with 4′,6-diamidino-2-phenylindole (DAPI) (blue).

**Figure 5 ijms-22-01062-f005:**
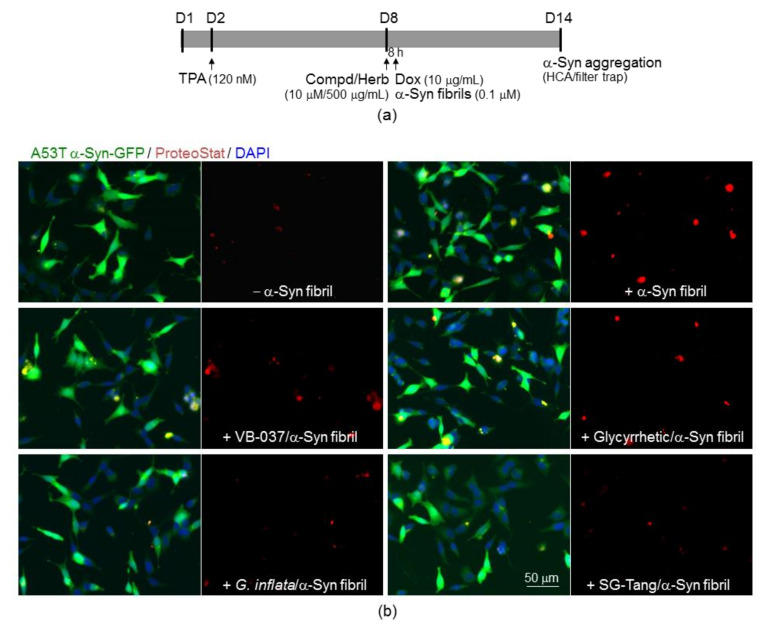
α-Synuclein aggregation analysis on A53T SNCA-GFP SH-SY5Y cells. (**a**) Experimental flow chart. Cells were seeded on day 1 (D1), with 12-O-tetradecanoylphorbol-13-acetate (TPA) (120 nM) added on day 2 (D2) to promote DAergic differentiation. On day 8 (D8), compound (10 µM) or herb (500 µg/mL) was added to the cells for 8 h, followed by induction of A53T SNCA-GFP expression with doxycycline (Dox; 10 µg/mL) and addition of preformed α-synuclein fibril (0.1 µM) for 6 days. On day 14 (D14), high content analysis (HCA) analysis of α-synuclein aggregation was performed using ProteoStat stained images. In addition, α-synuclein aggregates were measured by filter trap assay with a GFP antibody. (**b**) Fluorescent microscopy images of A53T SNCA-GFP-expressing cells (green) with or without preformed fibril addition, or VB-037, glycyrrhetic acid, *G. inflata* or SG-Tang treatment, with nuclei detected (blue) or aggregates marked (red). (**c**) HCA aggregation analysis of the A53T SNCA-GFP SH-SY5Y cells with VB-037, glycyrrhetic acid, *G. inflata* or SG-Tang treatment (*n* = 3). (**d**) Filter trap analysis of α-synuclein aggregates of the A53T SNCA-GFP SH-SY5Y cells with VB-037, glycyrrhetic acid, *G. inflata* or SG-Tang treatment. The α-synuclein aggregates were probed with anti-GFP antibody (*n* = 3). To normalize, the relative α-synuclein aggregates with fibril addition is set as 100%. *p* values: comparisons between with and without doxycycline addition (^###^: *p* < 0.001), between with and without fibril addition (^&&&^: *p* < 0.001), or between with and without compound/herb treatment (*: *p* < 0.05, **: *p* < 0.01 and ***: *p* < 0.001). (One-way ANOVA with a post hoc Tukey test).

**Figure 6 ijms-22-01062-f006:**
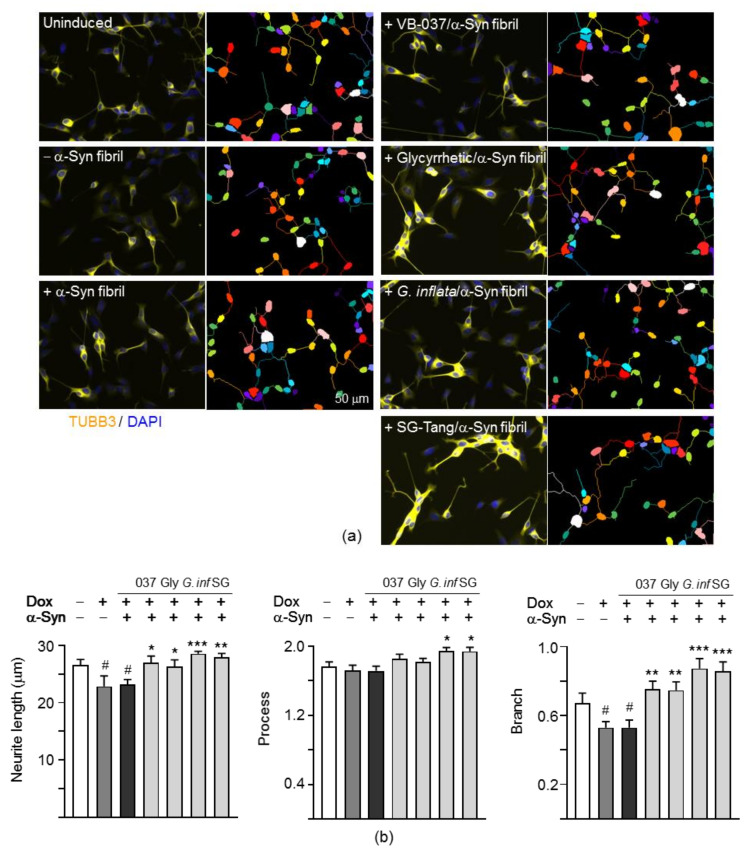
Neurite outgrowth and neuronal survival analyses on A53T SNCA-GFP SH-SY5Y cells. As described, TPA was added to the cells on day 2, and compound/herb, doxycycline and α-synuclein fibril were added on day 8. On day 14, neurite outgrowth, lactic dehydrogenase (LDH) release, reactive oxygen species (ROS) production and caspase 1/3 activities were measured. (**a**) Fluorescent microscopy images of A53T SNCA-GFP-expressing cells with or without preformed fibril addition, or VB-037, glycyrrhetic acid, *G. inflata* or SG-Tang treatment. Neuronal class III β-tubulin (TUBB3) staining was performed to quantify the extent of neurite outgrowth. Nuclei were detected with 4′,6-diamidino-2-phenylindole (DAPI) (blue). Segmented images with multi-colored masks to assign each outgrowth to a cell body for neurite outgrowth quantification were also shown. (**b**) Quantification of neurite length, brunch and process in A53T SNCA-GFP SH-SY5Y cells treated with VB-037, glycyrrhetic acid, *G. inflata* or SG-Tang (*n* = 3). (**c**) LDH release, ROS production and caspase 1/3 activities of A53T SNCA-GFP-expressing cells with or without preformed fibril addition, or VB-037, glycyrrhetic acid, *G. inflata* or SG-Tang treatment (*n* = 3). To normalize, the relative LDH/ROS/caspase 1/caspase 3 level in cells without doxycycline and α-synuclein fibril addition was set as 100%. *p* values: comparisons between with and without doxycycline addition (#: *p* < 0.05, ^##^: *p* < 0.01 and ^###^: *p* < 0.001), between with and without fibril addition (^&&&^: *p* < 0.001), or between with and without compound/herb treatment (*: *p* < 0.05, **: *p* < 0.01 and ***: *p* < 0.001). (One-way ANOVA with a post hoc Tukey test).

**Figure 7 ijms-22-01062-f007:**
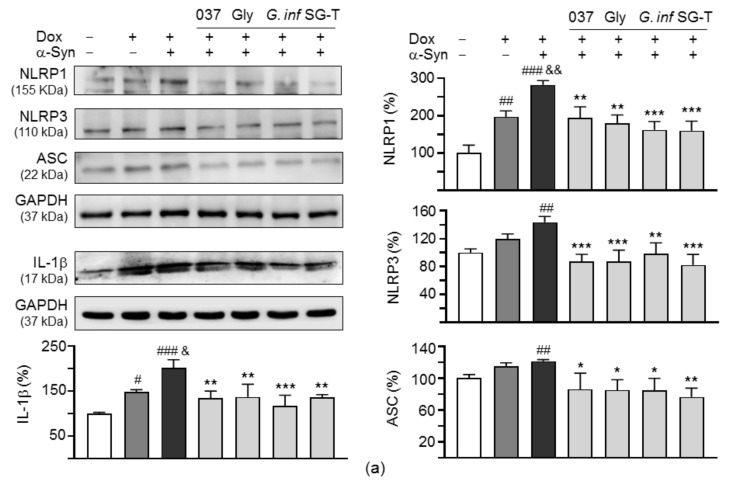
IL-1β- and IL-6-mediated signaling analyses on A53T SNCA-GFP SH-SY5Y cells. Western blot analysis of (**a**) NLR family pyrin domain containing 1 (NLRP1) and 3 (NLRP3), PYD and CARD domain containing (ASC), interleukin (IL)-1β, (**b**) c-Jun N-terminal kinase (JNK) (T183/Y185), proto-oncogene c-Jun (JUN) (S63), nuclear factor (NF)-κB inhibitor alpha (IκBα) (S32/36), NF-κB P65 subunit (P65) (S536), mitogen-activated protein kinase 14 (P38) (T180/Y182), signal transducer and activator of transcription 1 (STAT1) (S727), and (**c**) IL-6, Janus kinase 2 (JAK2) (Y1007/1008), signal transducer and activator of transcription 3 (STAT3) (Y705), suppressor of cytokine signaling 3 (SOCS3). Glyceraldehyde 3-phosphate dehydrogenase (GAPDH) was used as a loading control (*n* = 3). To normalize, the relative NLRP1, NLRP3, ASC, IL-1β, JNK, JUN, IκBα, P65, P38, STAT1, JAK2, STAT3, and SOCS3 of uninduced cells, or relative IL-6 of untreated cells was set at 100%. *p* values: comparisons between with and without doxycycline addition (^#^: *p* < 0.05, ^##^: *p* < 0.01 and ^###^: *p* < 0.001), between with and without fibril addition (^&^: *p* < 0.05, ^&&^: *p* < 0.01 and ^&&&^: *p* < 0.001), or between with and without compound/herb treatment (*: *p* < 0.05, **: *p* < 0.01 and ***: *p* < 0.001). (One-way ANOVA with a post hoc Tukey test)

**Figure 8 ijms-22-01062-f008:**
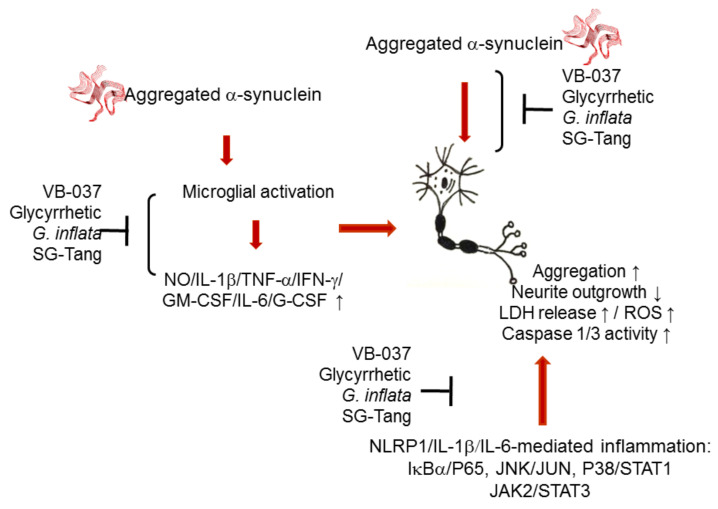
Graphical summary.

**Table 1 ijms-22-01062-t001:** Targets identified by mouse inflammation antibody array.

Cytokine	Fold Change (+ vs. - α-Syn)	Fold Change (+ VB-037/α-Syn vs. + α-Syn)	Fold Change (+ Glycyrrhetic/α-Syn vs. + α-Syn)
IL-1α	9.7	6.8	8.0
IL-1β	2.6	0.8	0.4
TNF-α	3.5	0.8	3.1
IFN-γ	1.7	1.2	1.0
GM-CSF	5.2	1.6	2.8
IL-6	104.9	65.9	73.7
G-CSF	97.7	42.9	54.0

## Data Availability

The data presented in this study are available on request from the corresponding author. The data are not publicly available due to privacy.

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
