# Peer review of "Pathomechanism Characterization and Potential Therapeutics Identification for Parkinson’s Disease Targeting Neuroinflammation"

_ijms, 2021, doi:10.3390/ijms22031062_

Round 1

Reviewer 1 Report

I think that it is a work that unites numerous experiments carried out conveniently and that one arrives at very interesting conclusions however some inconveniences exist.

In the 110 line, the process of inflammation is very complex and requires the intervention of numerous components, cytokine secretion is only one of them. I do not consider it appropriate to define this secretion or a model in vitro as inflammation. In any case, it could be defined as microglial activation but inflammation need many other compouds

In the line 119. Iba 1 is not an activated microglia marker

The overall assessment of microglial activation is very low. Cells in culture are not identified and morphological identification does not involve any semiquantitative evaluation at least, except for one wester. A confirmation with immunocytochemistry and probably with more specific markers of activated microglial

I do not know the components of the formulations based on SG-Tang and other plant extracts. The chemical composition or active principles and the doses are not named.

I  consider that an article should not be published if the chemical components that are making the effect in the herbal extracts are not clearly identified.

But I think it is a good article and could be reoriented

Thank you for your work

Author Response

Dear Editor,

Thank you for the review of the revised manuscript ijms-988887 entitled " Pathomechanism Characterization and Potential Therapeutics Identification for Parkinson’s Disease Targeting Neuroinflammation ". We thank the constructive comments and have revised our manuscript accordingly. The revised sections are highlighted in red font and listed below. Thank you for the prompt attention.

Yours sincerely,

Chiung-Mei Chen, M.D., Ph.D.

Response to Reviewer 1 Comments

Reviewer #1 (Comments and Suggestions for Authors):

I think that it is a work that unites numerous experiments carried out conveniently and that one arrives at very interesting conclusions however some inconveniences exist.

1. In the 110 line, the process of inflammation is very complex and requires the intervention of numerous components, cytokine secretion is only one of them. I do not consider it appropriate to define this secretion or a model in vitro as inflammation. In any case, it could be defined as microglial activation but inflammation need many other compounds.

Response: We have revised line 110 as suggested: 2.1. α-Synuclein Induced Microglial Activation in Mouse BV-2 Cells

2. In line 119, Iba1 is not an activated microglia marker. The overall assessment of microglial activation is very low. Cells in culture are not identified and morphological identification does not involve any semiquantitative evaluation at least, except for one western. A confirmation with immunocytochemistry and probably with more specific markers of activated microglia.

Response: We confirmed the α-synuclein-induced microglial activation by CD68 immunocytochemistry stain in Results (lines 120-121): Exposure of BV-2 cells to α-synuclein (2.5 µM) also resulted in increased expression of CD68 molecule (230%, p = 0.018) and MHCII (198%, p = 0.002) (Figure 1(e)).

Figure 1 legend (lines 132-135): (e) Immunocytochemistry of CD68 and MHCII (red) and fluorescent intensity with or without α-synuclein (2.5 μM) activation. Nuclei were counterstained with 4’,6-diamidino-2-phenylindole (DAPI) (blue). For normalization, the relative fluorescent intensity in inactive cells was set as 100% (n = 3).

In Materials and Methods, we added a paragraph to address the methods for detection of BV-2 activation (lines 461-472).

3. I do not know the components of the formulations based on SG Tang and other plant extracts. The chemical composition or active principles and the doses are not named. I consider that an article should not be published if the chemical components that are making the effect in the herbal extracts are not clearly identified.

Response: We added sentences to describe the active principle and dose for G. inflata and SG-Tang (lines 139-142): Ammonium glycyrrhizinate, a common active constituent in both G. inflata and SG-Tang, is a glycyrrhizic acid salt. It is hydrolyzed by intestinal flora to glycyrrhetic acid. The amounts of ammonium glycyrrhizinate in these two herbs were 2.23% (26.6 mM) in G. inflata extract [27] and 2.43% (14.52 mM) in formulated SG-Tang [25].

Reviewer 2 Report

In this manuscript Chiung-Mei Chen, et al. described a study of different aspects of involvement of neuroinflammation in Parkinson disease (PD) development and progression, and evaluation of protective effect of a number of preparations in two models of PD in vitro. Despite contribution of neuroinflammation in process of neurodegeneration in PD is well known, this field of investigation is still of current importance because many mechanisms of this contribution remain unexplored. The relevance of the search for new drugs for the treatment of PD is also beyond doubt.

The data of the work demonstrated significant reducing effect of the tested preparations on α-synuclein aggregation and associated oxidative stress that protected cells against α-synuclein-induced neurotoxicity. It is essential that the authors obtained results describing the molecular mechanisms of the registered effects. The results presented in the manuscript indicate that the tested drugs have real potential for the treatment of PD.

Manuscript is well written and the conclusions are convincingly supported by experimental results. Several minor suggestions will improve the overall quality of the manuscript:

  1. Line 134. It would be more correct to replace “IC50” by “IC50 for BV-2 viability”.
  2. Line 196. I think that abbreviation “TH” and “TPA” should be decoded here.
  3. Figure 6. (a) – the number of colors on the picture is more than two (not only blue and red). It should be commented.

Author Response

Dear Editor,

Thank you for the review of the revised manuscript ijms-988887 entitled " Pathomechanism Characterization and Potential Therapeutics Identification for Parkinson’s Disease Targeting Neuroinflammation ". We thank the constructive comments and have revised our manuscript accordingly. The revised sections are highlighted in red font and listed below. Thank you for the prompt attention.

Yours sincerely,

Chiung-Mei Chen, M.D, Ph.D

Response to Reviewer 2 Comments

Reviewer #2 (Comments and Suggestions for Authors):

In this manuscript, Chiung-Mei Chen et al. described a study of different aspects of involvement of neuroinflammation in Parkinson disease (PD) development and progression, and evaluation of protective effect of a number of preparations in two models of PD in vitro. Despite contribution of neuroinflammation in process of neurodegeneration in PD is well known, this field of investigation is still of current importance because many mechanisms of this contribution remain unexplored. The relevance of the search for new drugs for the treatment of PD is also beyond doubt. The data of the work demonstrated significant reducing effect of the tested preparations on α-synuclein aggregation and associated oxidative stress that protected cells against α-synuclein-induced neurotoxicity. It is essential that the authors obtained results describing the molecular mechanisms of the registered effects. The results presented in the manuscript indicate that the tested drugs have real potential for the treatment of PD.

The manuscript is well written and the conclusions are convincingly supported by experimental results. Several minor suggestions will improve the overall quality of the manuscript:

  1. Line 134. It would be more correct to replace “IC50” by “IC50 for BV-2 viability”.

Response: We replaced “IC50” with “IC50 for BV-2 viability” (line 143) as suggested.

  1. Line 196. I think that abbreviations “TH” and “TPA” should be decoded here.

Response: We defined “TH” and “TPA” in lines 207-208: Among them, clone 8 was further examined for DAergic neuronal marker tyrosine hydroxylase (TH) expression with 120 nM 12-O-tetradecanoylphorbol-13-acetate (TPA, also called phorbol 12-myristate 13-acetate) treatment (Figure 4(c)).

  1. Figure 6. (a) – the number of colors on the picture is more than two (not only blue and red). It should be commented.

Response: We explained the multi-colored images in Figure 6(a) (lines 292-293): Also shown were segmented images with the multi-colored mask to assign each outgrowth to a cell body for neurite outgrowth quantification.